Repeatability and reproducibility of Keratograph 5M corneal topography

Ortiz-Toquero Sara sara.ortiz@uva.es
Sanchez Irene
Martin Raul
1 Departamento de Física Teórica, Atómica y Óptica, Universidad de Valladolid , Valladolid , Castilla y León , Spain
2 Optometry Research Group, IOBA Eye Institute, University of Valladolid , Valladolid , Castilla y León , Spain
Gkantidis Nikolaos
Electronic publication date: 2025 May 22
Publication date: 2025
Volume: 13
Electronic Location ID: e19478
Received 2024 Oct 22; Accepted 2025 Apr 25
Copyright: ©2025 Ortiz-Toquero et al.
Copyright year: 2025
Copyright holder: Ortiz-Toquero et al.
License: This is an open access article distributed under the terms of the Creative Commons Attribution License, which permits unrestricted use, distribution, reproduction and adaptation in any medium and for any purpose provided that it is properly attributed. For attribution, the original author(s), title, publication source (PeerJ) and either DOI or URL of the article must be cited.
License URL: https://creativecommons.org/licenses/by/4.0/

Keywords: Corneal topography, Intrasession repeatability, Keratograph 5M, Intersession reproducibility, Placido disc, Test-retest, Corneal curvature, Corneal eccentricity, Corneal aberration, Contact lens

Funding: The authors received no funding for this work.

==============================
Background

Corneal topography is an important diagnostic tool and highly repeatable and reproducible topographic devices are essential in eye care practice. Placido disc-based topography is one of the most widely used methods because of its high resolution and accuracy. The aim of this study was to analyse the intrasession repeatability and intersession reproducibility of measurements obtained with a Keratograph 5M in a sample of healthy subjects.

Methods

Three consecutive measurements were performed with a Keratograph 5M during two sessions in 24 healthy subjects to calculate the within-subject standard deviation (Sw), repeatability and reproducibility limits, coefficient of variation (CoV), and intraclass correlation coefficient (ICC) of corneal curvature (K1, K2 and Max-K), eccentricity, corneal diameter, and corneal aberration (6-mm pupil; coma, trefoil, spherical aberration, secondary astigmatism and quadrafoil).

Results

No statistically significant differences were found between the three measurements in either session (P ≥ 0.06). The corneal parameters that demonstrated the best repeatability were corneal curvature and corneal diameter with a CoV, and the ICCs ranged from 0.41% and 0.990 (corneal diameter) to 0.28% and 0.998 (K2). Eccentricity and corneal aberrations had lower repeatability results, with CoVs and ICCs ranging from 3.88% and 0.992 (spherical aberration) to 40.21% and 0.643 (quadrafoil), respectively. In the case of reproducibility, excellent results were obtained for corneal curvature and diameter measurements (CoV ≤ 0.36% and ICC ≥ 0.987), with moderate reproducibility for corneal eccentricity (CoV ≥ 2.03% and ICC ≤ 0.986), secondary astigmatism (CoV = 20.05% and ICC = 0.787), and quadrafoil (CoV = 23.55% and ICC = 0.696).

Conclusions

The Keratograph 5M demonstrated excellent repeatability and reproducibility in measuring corneal curvature and corneal diameter in healthy subjects. Corneal eccentricity shows moderate accuracy, whereas corneal aberrations (except coma, trefoil, and spherical aberrations) exhibit moderate measurement reliability and should be interpreted with caution in clinical practice.

Introduction

Corneal topography is an important diagnostic tool in eye care practice, and it can provide detailed information on the corneal surface, which is crucial for many types of clinical decision-making (Goto & Maeda, 2020; Holladay, 2009; McAlinden, Khadka & Pesudovs, 2015). Anterior corneal surface assessment is essential in the diagnosis and management of several corneal conditions, such as keratoconus, for planning corneal refractive surgery or intraocular lens calculations, and in contact lens (CL) practice, particularly in gas permeable (GP) CL fitting (trial lens calculation, orthokeratology practice, irregular cornea patient management, etc.) (Goto & Maeda, 2020; Bhatoa, Hau & Ehrlich, 2010; Cavas-Martínez et al., 2016). Therefore, highly repeatable and reproducible topographic devices are essential for monitoring disease progression, ensuring accurate outcomes and assessing the effectiveness of treatments (corneal refractive surgery, orthokeratology, CL fitting, etc.) (McAlinden, Khadka & Pesudovs, 2015; McAlinden, Khadka & Pesudovs, 2011a; Aramberri et al., 2012).

Among the different techniques available to analyze the corneal shape (manual or automated keratometry, slit-scanning tomography, rotating Scheimpflug tomography or optical coherence tomography), Placido disc-based videokeratoscopy is one of the most widely used methods (Martin, 2018; Mülhaupt et al., 2018). This technique, first proposed by Klyce (1984), is the most frequently used corneal topography system in clinical practice because of its high resolution and accuracy (Martin, 2018). It involves imaging the anterior corneal surface by analyzing tear film reflections of multiple concentric rings projected onto the cornea. These devices capture reflected images from these rings allowing for precise analysis of anterior corneal shape and irregularities (Martin, 2018). There are several models on the market of videokeratoscopes based on Placido discs that have demonstrated excellent measurement accuracy (Hua et al., 2016; Wang et al., 2012; Mohamed et al., 2021).

The Keratograph 5M (Oculus, Wetzlar, Germany) is a Placido disc-based corneal topographer that is widely used in clinical practice due to its advanced imaging capabilities and comprehensive analysis features (García-Marqués et al., 2021). This device measures corneal curvature and provides additional data on tear film quality and meibomian gland function (García-Marqués et al., 2021). The repeatability of dry eye assessment tools (ocular redness, tear break-up time or tear meniscus height measurement) has been previously reported (García-Marqués et al., 2021; Alfaro-Juárez et al., 2019; Lee et al., 2016). However, to the best of the authors’ knowledge, the repeatability and reproducibility of corneal measurements provided by Keratograph 5M have not been reported. Assessing the consistency of corneal measurements obtained using the Keratograph 5M has become essential in both clinical and research conditions to confirm its ability to determine corneal curvature with a high degree of accuracy and reliability.

Therefore, the aim of this study was to analyse the intrasession repeatability and intersession reproducibility of corneal parameter measurements obtained with the Keratograph 5M topographer in a sample of healthy subjects to ensure that this device delivers reliable and reproducible outcomes for effective patient eye care in clinical and research practice.

Materials & Methods

Subjects

A comprehensive eye examination was conducted to ensure ocular health in healthy subjects aged between 18 and 35 years, with a best corrected visual acuity equal to or greater than 20/20. The exclusion criteria included prior eye surgery, a history of ocular disease, presence of corneal fluorescein staining (central or peripheral), and systemic or ocular conditions that affect the ocular surface. Only data from the right eye of each subject were included in the statistical analysis. The study adhered to the principles of the Declaration of Helsinki, and written informed consent was obtained from all participants following a detailed explanation of the study’s nature after receiving approval from the Human Sciences Ethics Committee of Valladolid Area-Este Clinic Hospital (PI 17-748).

Instrumentation

The Oculus Keratograph 5M employs 22 rings with a wavelength of 880 nm to project a pattern from a Placido disc onto the tear film surface. When these reflections are analysed, the Keratograph 5M generates detailed maps of the corneal surface. Additionally, this topographer utilizes different types of illumination for specific dry eye evaluations (infrared light for tear film measurement and blue light for fluorescein imaging).

Measurement procedure

Figure 1 shows the diagram of the study procedure. Two measurement sessions were scheduled one week apart at the same time (between 11 AM and 3 PM) to reduce the influence of diurnal variations on the anterior segment shape in both study sessions (Read & Collins, 2009). The subjects placed their chin on the chin rest and pressed their forehead against the forehead strap. The topographer mires were focused following the automated instructions provided by the instrument, and three consecutive measurements were taken from the undilated right eye of each participant. Measurements were performed after an eye blink, ensuring that the eye was aligned with the visual axis. To minimize the interdependence of successive measurements, the subjects removed their chin from the chin rest between scans. The same experienced operator conducted all the measurements following the same protocol and the guidelines provided by the manufacturer in a darkened room after verifying the instrument calibration. Data of poor quality due to factors such as eye movements, eyelid shadows, blinking, or artefacts from the tear film were discarded.

Figure 1 Flowchart of the study procedure.

The values of anterior corneal curvature (K1 and K2), curvature at the point of maximum power (Max-K), horizontal and vertical eccentricity, total eccentricity, corneal diameter, and corneal aberrations in the 6-mm pupil diameter (coma, trefoil, spherical aberration, secondary astigmatism, and quadrafoil) of axial curvature map were recorded during both sessions.

Statistical analysis

Statistical analysis was conducted with SPSS (version 26.0; IBM Corporation) and Microsoft Office Excel 365 (Microsoft Corp.). The Kolmogorov–Smirnov test was used to assess the normality of the data distribution, with P > 0.05 indicating a normal distribution. The estimated sample size calculation with a confidence level of 20% for three repeated measures was 24 subjects (McAlinden, Khadka & Pesudovs, 2015). This study followed the definitions of intrasession repeatability and intersession reproducibility according to the British Standards Institute and the International Organization for Standardization (McAlinden, Khadka & Pesudovs, 2011b).

To evaluate the intrasession repeatability of the Keratograph 5M, the following metrics were calculated and analysed for each session: mean ± standard deviation (SD); within-subject deviation (Sw) (Bland, 2000); the repeatability limit (r = 2.77 × Sw), which can be interpreted as the difference between two measurements of the same patient for 95% of pairs of observations (Bland, 2000), coefficient of variation (CoV), which was calculated as the ratio of Sw to the overall mean (Bland, 2000); and intraclass correlation coefficients (ICCs), ranging from 0 to 1, which are commonly classified as follows: <0.75 indicates poor agreement; 0.75 to <0.90 indicates moderate agreement; and ≥0.90 indicates high agreement (McGraw & Wong, 1996). Analysis of variance (ANOVA) for repeated measurements was used to compare the three measurements of both sessions for all study variables. The 95% limits of agreement (LoA) calculated as the mean difference ± 1.96 SD between measurements (Bland & Altman, 1986) and the exact 95% confidence intervals (Carkeet, 2015) were calculated following Bland–Altman analysis.

Intersession reproducibility was assessed via the mean of three repeat measurements from each session. The within-subject standard deviation (SR), reproducibility limit (R = 2.77 × SR), CoV, ICC, and LoA were also calculated (Bland, 2000; McGraw & Wong, 1996; Bland & Altman, 1986). The differences between the measurements of session 1 and session 2 were analysed with paired Student’s t tests (P < 0.05 was considered statistically significant).

Results

Patient demographics

This study involved 24 healthy volunteers (nine men and 15 women) with an average age of 21.95 ± 3.11 years (range 20–31 years) and a spherical equivalent refractive error of −2.35 ± 2.15 D (range −0.50 to −7.25 D).

Intrasession repeatability

The intrasession repeatability was excellent, with a CoV < 0.74% and an ICC ≥ 0.962 for the corneal curvature (K1, K2 and Max-K) and corneal diameter. The measurements of K1 (CoV ≤ 0.33% and ICC ≥ 0.996) and K2 (CoV ≤ 0.37% and ICC ≥ 0.996) showed the best agreement (Table 1). However, moderate agreement was found for the measurements of eccentricity (CoV ≥ 4.11% and ICC between 0.876 and 0.898) and corneal aberrations. Spherical aberration showed the best agreement (CoV ≥ 3.88% and ICC ≥ 0.986), whereas quadrafoil showed the worst results (CoV ≥ 36.02% and ICC ≤ 0.788), as summarized in Table 1.

Table 1 Intrasession repeatability of corneal measurements taken in sessions 1 and 2 by Keratograph 5M.

SD, standard deviation; Sw, intrasubject standard deviation; r, repeatability limit; CoV, coefficient of variation; ICC, intraclass correlation coefficient; LoA, limits of agreement.

Parameter	Session	Mean ± SD	Sw	r	CoV (%)	ICC	LoA	P Value†	
K1 (D)	1	43.23 ± 1.77	0.14	0.39	0.33	0.996	−0.61 to 0.45	0.09	
2	43.28 ± 1.71	0.12	0.33	0.28	0.997	−0.42 to 0.41	0.12	
K2 (D)	1	44.13 ± 1.94	0.13	0.37	0.30	0.996	−0.64 to 0.49	0.11	
2	44.14 ± 1.89	0.12	0.34	0.28	0.998	−0.40 to 0.42	0.23	
Max-K (D)	1	44.57 ± 1.99	0.23	0.64	0.52	0.994	−0.84 to 0.63	0.24	
2	44.68 ± 1.91	0.33	0.92	0.74	0.962	−1.82 to 1.73	0.46	
Horizontal eccentricity	1	0.55 ± 0.09	0.03	0.07	4.42	0.895	−0.16 to 0.13	0.40	
2	0.55 ± 0.09	0.02	0.05	4.11	0.898	−0.06 to 0.05	0.46	
Vertical eccentricity	1	0.57 ± 0.11	0.04	0.11	7.52	0.881	−0.22 to 0.15	0.06	
2	0.56 ± 0.12	0.03	0.08	6.91	0.898	−0.10 to 0.10	0.80	
Total eccentricity	1	0.56 ± 0.09	0.03	0.08	4.90	0.876	−0.18 to 0.14	0.30	
2	0.55 ± 0.10	0.02	0.06	4.12	0.889	−0.07 to 0.06	0.55	
Corneal diameter (mm)	1	11.78 ± 0.39	0.05	0.13	0.41	0.990	−0.16 to 0.22	0.24	
2	11.78 ± 0.39	0.05	0.13	0.39	0.992	−0.16 to 0.17	0.54	
Coma (μm)	1	0.18 ± 0.08	0.02	0.05	13.62	0.981	−0.05 to 0.06	0.06	
2	0.19 ± 0.08	0.02	0.06	14.41	0.969	−0.07 to 0.07	0.12	
Trefoil (μm)	1	0.12 ± 0.07	0.02	0.05	20.52	0.974	−0.07 to 0.05	0.11	
2	0.12 ± 0.07	0.02	0.06	24.37	0.947	−0.07 to 0.08	0.70	
Spherical aberration (μm)	1	0.23 ± 0.06	0.01	0.02	3.88	0.992	−0.02 to 0.03	0.70	
2	0.23 ± 0.06	0.01	0.03	5.90	0.986	−0.03 to 0.04	0.16	
Secondary astigmatism (μm)	1	0.04 ± 0.02	0.01	0.03	32.96	0.891	−0.03 to 0.03	0.65	
2	0.04 ± 0.02	0.01	0.03	35.77	0.877	−0.03 to 0.04	0.12	
Quadrafoil (μm)	1	0.05 ± 0.02	0.02	0.05	40.21	0.643	−0.05 to 0.05	0.46	
2	0.06 ± 0.03	0.02	0.05	36.02	0.788	−0.06 to 0.06	0.37	
Notes.

† ANOVA repeated test (P ≤ 0.05 statistically significant).

Intersession reproducibility

Table 2 shows the intersession reproducibility of the measurements between sessions 1 and 2. The corneal curvature (K1, K2, and Max-K) and corneal diameter parameters were the most reproducible, with CoVs less than 0.36% and ICCs ≥ 0.987, and K1 (CV = 0.21% and ICC = 0.998) was the parameter with better reproducibility (Fig. 2).

Table 2 Intersession reproducibility of corneal measurements taken in sessions 1 and 2 by Keratograph 5M.

SD, standard deviation; SR, intrasubject standard deviation; R, reproducibility limit; CoV, coefficient of variation; ICC, intraclass correlation coefficient; LoA, limits of agreement.

Parameter	Mean Diff ± SD	SR	R	CoV (%)	ICC	LoA	P Value †	
K1 (D)	−0.05 ± 0.15	0.09	0.24	0.21	0.998	−0.34 to 0.23	0.09	
K2 (D)	−0.01 ± 0.17	0.10	0.29	0.24	0.998	−0.34 to 0.32	0.81	
Max-K (D)	−0.03 ± 0.21	0.16	0.44	0.36	0.987	−0.44 to 0.38	0.22	
Horizontal eccentricity	0.00 ± 0.02	0.01	0.03	2.03	0.986	−0.04 to 0.04	0.93	
Vertical eccentricity	0.01 ± 0.05	0.03	0.08	5.51	0.944	−0.09 to 0.11	0.27	
Total eccentricity	0.01 ± 0.03	0.02	0.05	3.19	0.971	−0.06 to 0.07	0.45	
Corneal diameter (mm)	0.00 ± 0.07	0.04	0.10	0.32	0.992	−0.14 to 0.14	0.93	
Coma (μm)	0.00 ± 0.03	0.02	0.05	10.23	0.969	−0.06 to 0.05	0.53	
Trefoil (μm)	0.00 ± 0.04	0.02	0.06	16.15	0.920	−0.08 to 0.07	0.67	
Spherical aberration (μm)	0.00 ± 0.02	0.01	0.02	3.91	0.985	−0.03 to 0.03	0.89	
Secondary astigmatism (μm)	0.00 ± 0.02	0.01	0.02	20.05	0.787	−0.04 to 0.04	0.45	
Quadrafoil (μm)	−0.01 ± 0.02	0.01	0.04	23.55	0.696	−0.05 to 0.03	0.12	
Notes.

† Paired t test (P ≤ 0.05 statistically significant).

Figure 2 Bland-Altman plot for K1, K2, Max-K, and corneal diameter reproducibility.

The black solid lines represent the mean difference, and the dotted lines represent the LoA. The grey dotted lines show the 95% CI of LoA.

Similar to the intrasession repeatability results, both the corneal eccentricity (Fig. 3) and corneal aberrations (Fig. 4) showed the worst reproducibility results (Table 2) as ICC presented lower values (ICC ≤ 0.986) and CoV higher values (CoV ≥ 2.03%) compared to corneal curvature or corneal diameter. Similarly, the quadrafoil method had the lowest reproducibility coefficient (CoV = 23.55% and ICC = 0.696). Non-statistically significant differences (P ≥ 0.09) between the measurements taken in both sessions were found for all of the assessed parameters.

Figure 3 Bland–Altman plot for eccentricity reproducibility.

The black solid lines represent the mean difference, and the dotted lines represent the LoA. The grey dotted lines show the 95% CI of LoA.

Figure 4 Bland–Altman plot for corneal aberration reproducibility.

The black solid lines represent the mean difference, and the dotted lines represent the LoA. The grey dotted lines show the 95% CI of LoA.

Discussion

Assessment of the repeatability and reproducibility of corneal measurements obtained with any clinical device has become increasingly important in eye care practice for confirming the reliability and utility of tools. High-precision corneal measurements are essential in different clinical situations, particularly in preoperative planning for refractive surgery, in the diagnosis and monitoring of corneal pathologies such as keratoconus or in the prediction of lens parameters in regular and special contact lenses, such as the GP CL in irregular corneas or ortokeratology (Bhatoa, Hau & Ehrlich, 2010; Flynn et al., 2016). However, no previous reports have analysed the repeatability and reproducibility of corneal topographic parameters measured with the Keratograph 5M. Several publications have verified the reliability of this tool in analysing the tear film and other clinical signs of dry eye syndrome (Alfaro-Juárez et al., 2019; Baek, Doh & Chung, 2015) but not in analysing corneal parameters. In this study, the consistency of corneal curvature, corneal diameter, eccentricity, and corneal aberrations measured by Keratograph 5M across two sessions was analysed. The measurements of corneal curvature (K1, K2, and Max-K) and corneal diameter exhibited excellent repeatability (Fig. 2) and reproducibility (Table 2) in healthy subjects (CoV ≤ 0.74; ICCs ≥ 0.987). However, the measurements of corneal eccentricity (CoV ≤ 3.19; ICCs ≥ 0.876) and corneal aberrations (CoV ≤ 40.21; ICCs ≥ 0.643) were less accurate than the corneal curvature measurements. These findings can be attributed to the variance in the location of the pupil center in repeated measurements, which introduces noise and affects the repeatability of corneal aberrations, as noted by some authors, thus contributing to the overall variance (Wang, Shirayama & Koch, 2010; Cerviño et al., 2015). These results have implications for clinical decision-making processes that use corneal eccentricity (for example, calculating the optic zone base radius in GP CL fitting or reducing orthokeratology lens decentration (Gu et al., 2019; Li et al., 2017)) or biometric calculations in eyes postrefractive surgery, where corneal power cannot be measured with a manual keratometer (Piñero et al., 2017; Savini, Schiano-Lomoriello & Hoffer, 2018), or corneal aberrations (for example, in the early diagnosis of keratoconus (Ortiz-Toquero et al., 2016), postcorneal refractive surgery follow-up (Applegate et al., 2000)).

The repeatability of corneal curvature measurements is comparable to that previously reported with other available Placido-disk-based systems. Hua et al. (2016) analysed the precision of the Topcon KR-1W device (featuring 38 Placido rings) and reported very similar K1 (CoV = 0.25%, Sw = 0.16; ICC = 0.993) and K2 (CoV = 0.34%, Sw = 0.24; ICC = 0.993) repeatability to those reported in the present study. Additionally, the K1 intersession reproducibility (CoV = 0.21%, Sw = 0.20; ICC = 0.992) and K2 intersession reproducibility (CoV = 0.20%, Sw = 0.14; ICC = 0.993) were quite comparable to the results found with the Keratograph 5M in the current study (Hua et al., 2016). However, other Placido disc topographers (iTrace featuring 26 Placido rings) present slightly lower repeatability (CoV = 0.42%, Sw = 0.21; ICC = 0.993; K2: CoV = 0.50%, Sw = 0.26; ICC = 0.991) and reproducibility (CoV = 0.23%, Sw = 0.13; ICC = 0.996; K2: CoV = 0.30%, Sw = 0.17; ICC = 0.994) for K1 measurements (Hua et al., 2016).

Wang et al. (2012) observed excellent repeatability and reproducibility of corneal curvature (K1 and K2) measurements from three different Placido-disc-based devices (Medmont E300, EyeSys Vista, or Allegro Topolyzer). For the Medmont E300 (featuring 32 Placido rings), the CoV values and Sw were less than 0.18% and 0.08 D, respectively, with ICCs above 0.997. For EyeSys Vista (featuring 26 Placido rings), the CoV, Sw, and ICCs were less than 0.30%, less than 0.13 D, and above 0.989, respectively. For the Allegro Topolyzer (featuring 22 Placido rings), the CoV, Sw, and ICCs were less than 0.29%, less than 0.13 D, and higher than 0.993, respectively. Therefore, the intrasession repeatability and intersession reproducibility of the K1 and K2 measurements reported in the present study were quite similar to those reported for these three devices (Wang et al., 2012).

However, the Keratograph 5M has lower precision (Sw) in corneal curvature measurements than do Scheimpflug imaging-based topographers (Pentacam HR (Sw ≤ 0.07) (Kreps et al., 2020), Galilei (Sw = 0.07) (Mohamed et al., 2021), or Sirius (Sw ≤ 0.06) (Bayhan et al., 2014)) when the intrasession repeatability of corneal curvature (K1 and K2) is analysed in healthy subjects. This may suggest that the measurement of the curvature of the corneal anterior surface is more accurate with Scheimpflug technology.

The measurement of corneal eccentricity still presents challenges in terms of the precision of some current topographers, necessitating caution in their clinical application (Piñero et al., 2017; Ortiz-Toquero et al., 2014; Sanchez, Ortiz-Toquero & Martin, 2018). An older version of the Keratograph showed very similar results to those reported in this study (Ortiz-Toquero et al., 2014). Despite excellent repeatability for corneal curvature, lower repeatability for eccentricity values (CoVs of 5.79% and 14.53%) in healthy subjects and those with keratoconus has been reported. Similarly, the Visionix VX120 multidiagnostic unit also showed low repeatability for corneal eccentricity (Piñero et al., 2017; Sanchez, Ortiz-Toquero & Martin, 2018). As previously mentioned, the low repeatability of corneal eccentricity measurements has an impact on several procedures, such as orthokeratology, GP CL fitting, and biometric calculations, in postrefractive surgery patients (Piñero et al., 2017; Savini, Schiano-Lomoriello & Hoffer, 2018).

Corneal diameter measurement is commonly utilized in calculating the power and diameter of phakic or pseudophakic intraocular lenses, fitting the CL or diagnosing various corneal diseases, such as microcorneal diseases (Oleszko, Marek & Muzyka-Woźniak, 2021). The Keratograph 5M showed excellent repeatability and reproducibility (Sw ≤ 0.05; ICC ≥ 0.990) for measuring this parameter, which is comparable with previous reports in the adult population with other devices, such as the IOL Master 700 (Sw = 0.05 mm; ICC = 0.990), Galilei G2 (Sw = 0.03 mm; ICC = 0.995), or DRI OCT Triton (Sw = 0.06 mm; ICC = 0.993) (Boyle et al., 2021).

Additionally, the analysis of corneal aberrations is used in several clinical situations, such as in the prediction of visual performance, early detection of keratoconus, and evaluation of postoperative outcomes of laser refractive surgery (Applegate et al., 2000). Aberration terms, such as coma and spherical aberration, effectively describe decentration and surface irregularities resulting from laser ablation complications (Baek, Doh & Chung, 2015) or corneal conditions such as keratoconus (Ortiz-Toquero et al., 2016). The Keratograph 5M showed good precision in the measurements for coma, trefoil, and spherical aberrations (ICCs ≥ 0.920), moderate agreement for secondary astigmatism (ICCs ≥ 0.787), and poor agreement for quadrafoil (ICCs ≥ 0.643). These results are consistent with those of Piñero et al. (2017) who also reported worse repeatability for quadrafoil (Sw = 0.025; ICC = 0.689) and secondary astigmatism (Sw = 0.014; ICC = 0.887) measurements with the VX120 device and slightly better repeatability for coma (Sw = 0.031; ICC = 0.916), quadrafoil (Sw = 0.036; ICC = 0.845), and spherical aberration (Sw = 0.021; ICC = 0.958) measurements (Bland & Altman, 1986) with the Keratograph 5M. Bayhan et al. (2014) found a relatively lower degree of repeatability in anterior corneal aberrometric measurements with the Sirius system in healthy eyes, with ICCs varying from 0.568 for quadrafoil to 0.856 for coma. Shetty et al. (2022) reported better repeatability results for coma with the AXL Wave (CoV = 8.9%) and iTrace (CoV = 9.2%) devices and poorer results for spherical aberrations (CoV = 8.6% and 9.1%, respectively). The Galilei G4 tomographer also showed poorer repeatability for aberration measurements, with CoV and ICC values of 53% and 0.669, respectively, for quadrafoil aberrations and 7% and 0.981, respectively, for spherical aberrations (Wang, Shirayama & Koch, 2010). In general, the agreement with respect to corneal aberration measurements in healthy subjects is consistent with previously published results with other devices (Piñero et al., 2017; Bayhan et al., 2014; Wang, Shirayama & Koch, 2010). The repeatability of corneal aberration measurements is lower in healthy subjects with regular corneas than in those with irregular corneas (Ortiz-Toquero et al., 2016). This could be because corneal aberrations are greater and clearly defined with increasing corneal irregularity, so their measurement could be more repeatable and useful for disease classification.

While the Keratograph 5M has demonstrated robust accuracy in general, it is important to acknowledge potential limitations. Factors such as patient cooperation, ocular surface irregularities, and environmental conditions during measurements can influence the consistency of the data. Only healthy eyes without any pathology were included, so it is possible that in eyes with corneal abnormalities, the precision results may differ. The differences between the three measurements in both sessions are close to statistical significance for K1 or coma, could be attributed to the relatively small number of eyes examined. Future studies focusing on minimizing these variables or exploring their effects on device repeatability are needed. Additionally, further research could investigate the repeatability of the Keratograph 5M in unique populations, such as paediatric, keratoconus, or postsurgical patients, to ensure its broader applicability.

Conclusions

The Keratograph 5M shows excellent repeatability and reproducibility in measuring corneal curvature and corneal diameter in healthy subjects, making it a reliable tool in both clinical and research settings. Its high level of consistency supports its use in critical areas such as preoperative assessment, diagnosis, and long-term monitoring of corneal conditions.

The findings from this study allow to affirm role of the Keratograph 5M as a key instrument in eye care clinical practice while also highlighting areas for future research to further enhance its utility. Additionally, corneal eccentricity shows moderate accuracy, whereas corneal aberrations (except coma, trefoil, and spherical aberration) show moderate measurement reliability in healthy eyes without pathology. Therefore, corneal eccentricity, secondary astigmatism, and quadrafoil values should be interpreted with caution in clinical practice.

Supplemental Information

Supplemental Information 1 Breakdown of data from both visits for all patients

Additional Information and Declarations

Competing Interests

Author Contributions

Human Ethics

Data Availability

The authors declare there are no competing interests.

Sara Ortiz-Toquero conceived and designed the experiments, performed the experiments, analyzed the data, prepared figures and/or tables, authored or reviewed drafts of the article, and approved the final draft.

Irene Sanchez conceived and designed the experiments, performed the experiments, authored or reviewed drafts of the article, and approved the final draft.

Raul Martin conceived and designed the experiments, analyzed the data, prepared figures and/or tables, authored or reviewed drafts of the article, and approved the final draft.

The following information was supplied relating to ethical approvals (i.e., approving body and any reference numbers):

The study received the approval from the Human Sciences Ethics Committee of Valladolid Area-Este Clinic Hospital (PI 17-748).

The following information was supplied regarding data availability:

The raw data is available in the Supplemental File.

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
