# Peer review of "Repeatability and reproducibility of Keratograph 5M corneal topography"

_PeerJ, doi:10.7717/peerj.19478_

## Round 0.1 · original submission · Major Revisions

Please respond to the comments of all three reviewers.

·

Basic reporting

The authors tested the repeatability and reproducibility of the Keratograph 5M topographer on young, healthy subjects. They found good repeatability and reproducibility for most of the major clinical metrics; however, the results for some advanced metrics (e.g., eccentricity and higher order aberrations) were not as strong. The manuscript is generally well written, but the authors need to address several framing and methodological issues. See below.

In the Introduction, the authors should clarify that corneal topography analyzes the anterior surface of the corneal only, in contrast to corneal tomography, which analyzes the anterior and posterior surfaces (lines 58-59).

In the Discussion, the authors should provide rationale 1) for why some outcome metrics had superior repeatability and reproducibility than others within the device (lines 179-182) and 2) for why corneal tomography measurements are more precise than corneal topography measurements (lines 201-213).

A citation is needed to support the claim that corneal parameters are subject to diurnal fluctuations (lines 94-95).

The numeric values provided in the captions for all three figures are unnecessary/redundant because they can be gathered from the plots themselves. At the very least, they should be moved to the main body of the text, if not deleted completely.

In the Figures, the authors should be more specific in labeling the axes. Difference between what (y-axis)? Mean of what (x-axis)?

Experimental design

The authors never make an argument for the need for this study. They claim in in the Introduction that “the repeatability and reproducibility of corneal topography have not been reported” (lines 66-67). However, a quick literature search revealed many such studies (e.g., Guven, 2022; Xu et al., 2016; Lee et al., 2018; Doganterol et al., 2018; Ortiz-Toruero et al., 2014; Angelo et al., 2023, etc). Moreover, the authors spend much of the Discussion comparing their results to those of previous studies, negating this claim in the Introduction. Thus, they need to reframe their Introduction to better justify their study. What gap is in the literature, and why is their study needed?

Is any corneal staining an exclusion criterion? What about a single small punctate erosion in the peripheral cornea? The authors need to clarify this issue (Line 79).

Were the outcome metrics based on a tangential or axial algorithm? This is a critical detail as the two are not identical (see Yuhas et al., 2024). If data from both algorithms are available, are the results consistent between the two?

Validity of the findings

Although not pertinent to the results of this study, the authors should be wary of running multiple statistical analyses without correcting their p-value for multiple comparisons.

Additional comments

In the abstract, it would be helpful to have a separate section for the conclusion, instead of lumping it together with the results.

Reviewer 2 ·

Basic reporting

Dear authors,

I would like to thank you for your work.
The study is well written, has clear methodology and appropriate use of statistical methods.
Hereby my comment on your work.
The fact that the differences between the three measurements are close to statistical significance for K1 and several of the abberations, could be attributed to the relatively small number of eyes examined in your study, this should be explained in your study as a limitation.

Experimental design

The study design is well presented and clear.

Validity of the findings

No comment.

Reviewer 3 ·

Basic reporting

The language of this paper is well written. For the introduction There is a lack in the literature review (you have to add more papers related to your study). The structure of this paper well made. I suggest making a diagram to present the steps of the study for it is not so clear.

Experimental design

- The 24 healthy subjects are a small number to make a study on. I suggest to increase the number of the studied subjects.
- Have you tried to involve unhealthy subject to the study.
- In line (79-80) you mentioned ‘’ only data 80 from the right eye of each subject were included’’ have you tried to include the left eye or make a comparison between the results from the right eyes and the left eyes?
- In line (148-149) you mentioned ‘’ , moderate agreement was found for the measurements of eccentricity 149 (covg4.11% and icc between 0.876 and 0.898) and corneal aberrations.’’ can you explain why it showed moderate agreement whereas . The measurements of k1 147 (covf0.33% and iccg0.996) and k2 (covf0.37% and iccg0.996) showed the best agreement?
- You have to explain the terms that has been used in your paper
- Line (159-160) ‘’ , both the corneal eccentricity (figure 2) and 160 corneal aberrations (figure 3) showed the worst reproducibility results’’ can you explain why?
- In conclusion in lines (267-268) ‘’ the keratograph 5m shows excellent repeatability and reproducibility in measuring corneal 268 curvature and corneal diameter in healthy subjects’' have tried another device to say that?

Validity of the findings

- In line (66-67) you mentioned that’’ the repeatability and reproducibility of corneal topography measurements have not been reported.’’
but these are some papers that have studied this part:
a- Reproducibility, and repeatability of corneal topography measured by revo nx, galilei g6 and casia 2 in normal eyes in 2020
b- Precision (repeatability and reproducibility) and agreement of corneal power measurements obtained by topcon kr-1w and itrace 2016
c- Cepeatability, reproducibility, and agreement characteristics of rotating scheimpflug photography and scanning‐slit corneal topography for corneal power measurement 2009.

Additional comments

- In (278) there is a typo.
- In lines (129 , 184 and 211) the reference was written in a wrong way.

---

## Round 0.2 · accepted · Accept

All reviewers' comments have been adequately addressed, and the manuscript can be published in its current form following careful editing.

·

Basic reporting

The authors addressed adequately all of my comments from the initial version of the paper.

Experimental design

The authors addressed adequately all of my comments from the initial version of the paper.

Validity of the findings

The authors addressed adequately all of my comments from the initial version of the paper.

Additional comments

The authors addressed adequately all of my comments from the initial version of the paper.

Reviewer 3 ·

Basic reporting

The article is well written and clear and the author added more articles that related to the subject in study. the structure of the article is much more clear and well structured the author took the advice of making a diagram for the work.

Experimental design

The author answered all the questions.

Validity of the findings

The conclusions are well stated and more clear .